# Mechanical Behaviour of Plantar Adipose Tissue: From Experimental Tests to Constitutive Analysis

**DOI:** 10.3390/bioengineering11010042

**Published:** 2023-12-31

**Authors:** Sofia Pettenuzzo, Elisa Belluzzi, Assunta Pozzuoli, Veronica Macchi, Andrea Porzionato, Rafael Boscolo-Berto, Pietro Ruggieri, Alice Berardo, Emanuele Luigi Carniel, Chiara Giulia Fontanella

**Affiliations:** 1Department of Civil, Environmental and Architectural Engineering, University of Padova, 35131 Padova, Italy; sofia.pettenuzzo@unipd.it (S.P.); alice.berardo@unipd.it (A.B.); 2Musculoskeletal Pathology and Oncology Laboratory, Department of Surgery, Oncology and Gastroenterology, University of Padova (DiSCOG), Via Giustiniani 3, 35128 Padova, Italy; elisa.belluzzi@unipd.it (E.B.); assunta.pozzuoli@unipd.it (A.P.); 3Orthopedics and Orthopedic Oncology, Department of Surgery, Oncology and Gastroenterology (DiSCOG), University of Padova, Via Giustiniani 3, 35128 Padova, Italy; pietro.ruggieri@unipd.it; 4Centre for Mechanics of Biological Materials, University of Padova, 35131 Padova, Italy; veronica.macchi@unipd.it (V.M.); andrea.porzionato@unipd.it (A.P.); rafael.boscoloberto@unipd.it (R.B.-B.); emanueleluigi.carniel@unipd.it (E.L.C.); 5Department of Neuroscience, Institute of Human Anatomy, University of Padova, 35121 Padova, Italy; 6Veneto Region Reference Center for the Preservation and Use of Gifted Bodies, Veneto Region, 35121 Padua, Italy; 7Department of Biomedical Sciences, University of Padova, 35131 Padova, Italy; 8Department of Industrial Engineering, University of Padova, 35131 Padova, Italy

**Keywords:** plantar adipose tissue, foot, biomechanical behaviour, unconfined compression tests, indentation tests, constitutive modelling, finite element modelling

## Abstract

Plantar adipose tissue is a connective tissue whose structural configuration changes according to the foot region (rare or forefoot) and is related to its mechanical role, providing a damping system able to adsorb foot impact and bear the body weight. Considering this, the present work aims at fully describing the plantar adipose tissue’s behaviour and developing a proper constitutive formulation. Unconfined compression tests and indentation tests have been performed on samples harvested from human donors and cadavers. Experimental results provided the initial/final elastic modulus for each specimen and assessed the non-linear and time-dependent behaviour of the tissue. The different foot regions were investigated, and the main differences were observed when comparing the elastic moduli, especially the final elastic ones. It resulted in a higher level for the medial region (89 ± 77 MPa) compared to the others (from 51 ± 29 MPa for the heel pad to 11 ± 7 for the metatarsal). Finally, results have been used to define a visco-hyperelastic constitutive model, whose hyperelastic component, which describes tissue non-linear behaviour, was described using an Ogden formulation. The identified and validated tissue constitutive parameters could serve, in the early future, for the computational model of the healthy foot.

## 1. Introduction

Plantar soft tissue is a composite tissue located between foot bony segments, plantar fascia, and skin. It is mainly formed by adipose chambers surrounded by connective septa (composed of collagen and elastin), whose dimensions and orientation change according to the foot location, i.e., rare or forefoot. From a histological point of view, the heel pad tissue’s structure is hierarchically organized according to a honeycomb configuration. Here, the fibrous tissue strands linking the skin dermis and the calcaneum define circular or cone-shaped septa [1,2]. The forefoot fat pad shows a similar organization to that of the rear region, with a more complicated structural conformation due to the metatarsal arrangement [3,4].

The closed-cell configuration of plantar soft tissue is related to its mechanical role, which is to withstand pressure loads and provide a damping system able to adsorb foot impact and shock phenomena. Its configuration and location also contribute to transfer loads to the structure of the foot during the phases of the gait or running cycle and to bear the body weight during static standing [5,6].

Plantar tissue degeneration, which can be due to different pathologies or the ageing of the subject [7], may compromise the mechanical functionality of the whole foot. One of these pathologies is metatarsalgia, which affects the metatarsal region of the foot and is usually caused by excessive sport activity [8,9]. Foot tissue degeneration can also occur in patients affected by diabetes, a systemic disorder that can lead to the development of foot ulcers that often require surgery [10,11,12]. Therefore, a better understanding of the biomechanical properties of the plantar soft tissue is of great interest to identify the most suitable treatment [13] and the more appropriate orthosis [14].

In recent years, computational models of the foot have been developed to analyse the phenomena that occur inside it, in relation to quantities that are non-obtainable through in vivo studies, such as stress and strain fields [15,16]. To develop a realistic computational model, a proper constitutive formulation of the tissues is necessary, which needs experimental results to be assessed. Different in vivo and in vitro techniques have been used to investigate the mechanical properties of the plantar adipose tissue. Regarding the experimental methods, impact tests [17] and indentation tests [18,19] were mainly used, but many of them focused only on a specific foot region, such as the heel pad [18,20] or the metatarsal one [17,21]. One of the main issues of these methodologies is the lack of assurance if the results of these tests are representative of the material properties of the fat pad tissue only or of other foot structures, such as plantar fascia and bones. In vitro tests overcome this issue, testing the plantar adipose tissue through unconfined compression tests [22,23], indentation tests, and shear tests [24,25], in both healthy and pathological tissues, such as diabetic ones [26,27]. The results of these tests are more reliable than those obtained from the in vivo ones. First of all, this is because just the plantar adipose tissue was tested, then because both the heel pad and lateral and metatarsal regions were investigated [23,27]. Nevertheless, these studies were performed only on samples harvested from cadaveric feet and none of them tested the adipose tissue from the medial region of the foot. The results obtained from the experimental tests were used to define the constitutive model, to determine its parameters, and to validate them. Some studies adopted a visco-hyperelastic model describing only the heel pad tissue’s parameters [22], while others used a quasi-linear viscoelastic model (QLV, described by Fung [28]) to describe the mechanical behaviour of the different foot regions, not considering the hyperelastic phenomena that occur inside the tissue [23,24,27].

In this context, the aim of this study is to fully describe the mechanical behaviour of the plantar adipose tissue, overcoming all the issues previously mentioned, by performing multiple experimental tests on samples harvested from both human donors and cadavers and taken from all foot regions (including the medial one). Furthermore, an incompressible visco-hyperelastic constitutive model was formulated, and the parameters were identified through an inverse Finite Element Analysis (FEA) of experimental results and finally validated. The most innovative thing in this work will be the complete mechanical characterization of the whole plantar adipose tissue of a healthy subject, leading to a set of validated constitutive parameters, that can be directly used and introduced in the finite element (FE) model of the foot.

## 2. Materials and Methods

### 2.1. Sample Collection and Dissection

Plantar adipose samples were collected from patients who underwent amputation for cancer at the University-Hospital of Padova (CESC Code: AOP2649) and from gifted bodies, according to Body Donation Program of the Institute of Anatomy of the University of Padova [29]. A total of seven subjects were enrolled for this study and their demographic and clinical data are reported in Table 1.

Following anatomical dissection on amputated lower limbs or gifted bodies, the plantar foot was delivered to the Laboratory of Mechanics of Biological Tissues of the University of Padova and stored at −20 °C in sealed vials. The plantar fat pad was removed from the surrounding tissues (plantar fascia and tendons) and adipose samples were cut from all foot regions, precisely from the heel pad, lateral, medial, and metatarsal regions (Figure 1a), and were dissected from the underlying skin (Figure 1b). Then, samples were washed with cold saline to eliminate tissue debris and blood residuals. After this procedure, they were manually trimmed into prismatic samples using a scalpel (for samples undergoing compression tests) or cut into cylindrical samples using a 20 mm punch (for samples tested with an indentation procedure) (Figure 1c,d). The size of the samples was measured with a manual calliper and was approximately 7.4 ± 1.7 mm for the height and 20.1 ± 1.1 mm for the side regarding cubic samples and it was around 10.6 ± 3.2 mm for the cylindric samples’ height. A total of 109 samples were obtained from the different foot regions: 47 from the heel pad, 25 from the lateral region, 9 from the medial region, and 28 from the metatarsal region. All the mechanical tests were performed at room temperature within a few hours of thawing. It was observed that this storage method does not significantly alter the mechanical properties of soft biological tissues [30].

### 2.2. Mechanical Tests

Unconfined compression tests on plantar adipose samples were performed with the Biomomentum testing machine (Model Mach-1 v500css, ©Biomomentum Inc., Laval, QC, Canada) using the uniaxial load cell with a capacity of 250 N and resolution of 0.0125 N. The material testing machine was equipped with a flat indenter, directly screwed into the load cell, with a diameter of 31.75 mm (Figure 2a) to realize the compression experiments, and with a plate above which the sample was positioned for the test. Both the flat indenter and the plate were in stainless steel 316 L material. Regarding the indentation test, a Galdabini material testing machine (Galdabini SpA, Varese, Italy) was used and adequately equipped with a spherical indenter of a 15 mm diameter, fixed between the machine’s grips, and a load cell with capacity of 25 kN and accuracy of 0.1% (Figure 2b). This set-up was the same as that adopted in two previous works [31,32]. Concerning the spherical indenter, it was realized through 3D printing, and it was in plastic material, as well as the cylinder used to contain the sample during the testing procedure.

During preparation and testing, samples were continuously moistened with a saline solution at room temperature. They were placed directly between the smooth platen and the flat indenter of the testing machine without using either sandpaper or glue to prevent sample slip during unconfined compression testing, while they were placed in a cylindrical container of 20 mm in diameter for indentation tests. Specimens were tested adopting unconfined compression and indentation procedures, aiming to identify plantar adipose samples’ non-linear and time-dependent response. All regions of the foot were tested considering the different procedures, with the exception of the medial region, which was not investigated with the indentation tests due to its low thickness.

Sample preconditioning was applied for both types of tests. Each sample was preconditioned by applying 5 loading–unloading cycles with strain equal to 8% at 10%/strain rate [33].

#### 2.2.1. Unconfined Compression Tests

Unconfined compression tests at three different strain rates were carried out on 38 samples harvested from patients who underwent amputation, leading the sample to 50% strain. For each strain rate, five cycles were performed, and the three strain rates adopted were 7%/s, 70%/s, and 700%/s [23]. To allow complete recovery of the tissue, five minutes of rest was adopted between tests at different strain rates (Figure 3a,b) [15].

To evaluate the viscoelastic behaviour of the tissue, a stress relaxation test composed of 6 subsequent steps was performed on 40 samples obtained from patients who underwent amputation, and each step was composed of 10% compression strain at a 100%/s strain rate (Figure 3c,d). A subsequent 300 s of resting time was imposed to allow the almost complete development of relaxation phenomena [25,27,31,32], leading the sample to a 60% compression strain at the end of the sixth relaxation step.

#### 2.2.2. Indentation Tests

After sample preconditioning, 5 stress relaxation ramps were performed on 31 samples obtained from human cadavers, each one constituted of 15% indentation strain (as the ratio between the indenter displacement and the initial thickness of each sample) at a 3000%/s strain rate. Then, 300 s of resting time was imposed, leading the adipose tissue to a 75% indentation strain at the end of the fifth step.

### 2.3. Data Elaboration

Force–position results and force–time data, obtained from tests at different strain rates and stress relaxation procedures, respectively, were analysed using Matlab (MATLAB R2021, The MathWorks Inc., Natick, MA, USA). Stress data were obtained dividing the force results by the sample area, which was computed through a picture of each sample taken before testing.

Concerning unconfined compression tests performed at different strain rates, stress–strain curves from different foot regions were compared at a specific velocity, and stress–strain curves within a specific foot region were analysed to evaluate the influence of strain rates. The same procedure was adopted to evaluate the initial (0–25% compression strain range) and final (47–50% compression strain range) compression elastic modulus (or Young’s modulus) of the above-mentioned stress–strain curves.

Regarding the stress relaxation protocol performed with the unconfined compression tests, stress and compression strain data at the end of the resting stages led to stress–strain curves at equilibrium (Figure 3d,e). The equilibrium stress–strain curves define the sample behaviour when all the time-dependent micro-structural rearrangement phenomena occur during testing [34]. Stress relaxation data were processed to identify the drop of normalized stress with time, where the normalized stress was defined as the ratio between the current stress and the peak stress of each rest phase. The equilibrium stress–strain and normalized stress–time curves from each sample were interpolated by typical exponential formulations using Matlab curve fitting tools (growing and decreasing exponential function for stress–strain and stress–time curves, respectively) to provide a continuous set of data and to compare the results at the same strain/time conditions. Finally, the initial and final elastic modulus of equilibrium stress–strain curves were calculated as the slopes of the curve in the toe (0–15% compression strain range) and the quasi-linear (46–50% compression strain range) regions, respectively. As for the stress relaxation tests carried out with the indentation procedure, an equal data elaboration was performed. In this case, there were force and indentation strain data instead of stress and compression strain ones; the initial and final stiffness of equilibrium force–indentation strain curves were calculated as the slopes of the curve in the toe (0–15% indentation strain range) and the quasi-linear (46–50% indentation strain range) regions, respectively.

### 2.4. Constitutive Model

The experimental results obtained from unconfined compression tests performed at different strain rates and stress relaxation tests showed a non-linear time-dependent response of the plantar adipose tissue, and the histomorphometric analysis of the histological conformation of the adipose microstructure (micro- and macro-chambers) suggested an isotropic and almost incompressible behaviour [15,35]. Consequently, a visco-hyperelastic formulation was adopted to describe the mechanical response of plantar adipose tissue. The mathematical formulation was based on the definition of the Helmholtz free energy *ψ*, as a function of the right Cauchy–Green strain tensor ***C*** and the viscous variables ***q^i^*** [15,28]:
(1)
$$\psi \left(C,q^{i}\right)=W^{\infty }\left(C\right)+\sum_{i=1}^{n}\int_{0}^{t}\frac{1}{2}q^{i}\left(s\right):\dot{C}ds$$

where *W*^∞^(***C***) is the hyperelastic potential associated with the equilibrium response of the material, while the second term defines the viscous contribution. The hyperelastic formulation is described using an Ogden model:
(2)
$$W^{\infty }\left({\tilde{\lambda }}_{1},\;{\tilde{\lambda }}_{2},\;{\tilde{\lambda }}_{3}\right)=\frac{2µ}{\alpha^{2}}\left({{\tilde{\lambda }}_{1}}^{\alpha }+{{\tilde{\lambda }}_{2}}^{\alpha }+{{\tilde{\lambda }}_{3}}^{\alpha }-3\right)+\frac{1}{D}{\left(J-1\right)}^{2}$$

where 
${\tilde{\lambda }}_{i}$
 are the deviatoric principal stretches with 
${\tilde{\lambda }}_{i}=J^{-\frac{1}{3}}\lambda_{i}$
, *λ_i_* represents the principal stretches, 
$J=\sqrt{det\left(C\right)}$
 is the deformation Jacobian, and µ, *α*, and *D* are the hyperelastic parameters. µ defines the shear modulus, *α* describes the non-linearity of tissue elasticity, as the increase in the elastic stiffness with stretching phenomena, and *D* represents the compressibility parameters. It was decided to adopt the Ogden model since it was best suited to represent soft isotropic biological tissue, like the adipose one [36]. The viscous variables ***q^i^*** are associated with the set of *n* viscous processes, chosen as equal to two in this formulation, that develop within the material under loading, and their evolution is defined by means of integral formulations that depend on the stress–strain history [28]:
(3)
$$q^{i}\left(t\right)=\frac{\gamma^{i}}{\gamma^{\infty }\tau^{i}}\int_{0}^{t}exp\left(-\frac{t-s}{\tau^{i}}\right)\left(2\frac{𝜕W^{\infty }(C)}{𝜕C}\right)ds$$

where *τ^i^* are the relaxation times, which measure the time taken to develop the viscous phenomena, *γ^i^* are the relative stiffnesses that define the contribution of stiffness to the viscous processes, and *γ*^∞^ = 1 − 
$\sum_{i=1}^{n}\gamma^{i}$
 is the equilibrium relative stiffness.

### 2.5. Computational Analyses

Constitutive parameters were identified and validated using the inverse analysis of data from the mechanical tests, using finite element models mimicking the specific testing procedures. Specifically, the equilibrium stress–strain curves were adopted to identify the hyperelastic constitutive parameters for different foot regions, while other experimental tests were used for validation. The geometrical model used to mimic the unconfined compression test was developed within the FE pre-processing software Abaqus CAE 2022 (Dassault Systèmes Simulia Corp., Providence, RI, USA). The plantar adipose sample was modelled as a prismatic body with a 7.4 mm height and 20 mm on the side, which represents an average adipose sample, while the flat indenter and the support plate were represented as cylinders with a 31.75 mm diameter and with a height of 3 mm and 5 mm, respectively. A rigid body formulation was assumed for the flat indenter and the plate. A penalty coefficient of 0.01 (almost frictionless [37,38]) and hard contact were imposed between the sample and the machine components to describe the interface among them. Linear 8-node hexahedral elements (C3D8RH) were used to discretize all the model components. The mesh seed of 1 mm and 0.7 mm for the machine components and the adipose samples, respectively, were chosen to obtain a good compromise between the accuracy of the results and computational cost. This led to a final configuration of about a 27,000-node model. Boundary conditions were applied to mimic what happens during mechanical tests. For this reason, the plate was fixed along the x-y-z-axis at the bottom side, while the flat indenter was fixed along the x- and z-axis. A vertical displacement, along the y-axis, was applied to the indenter towards the tissue model adopting a static analysis. Stress–strain at equilibrium and stress relaxation conditions were evaluated through numerical simulations performed with the computational model. The adipose tissue was described with the isotropic, almost incompressible, visco-hyperelastic constitutive model reported in the previous section, and the constitutive parameters were identified by the inverse FEA, comparing experimental data and model results using a cost function. More precisely, the optimal constitutive parameters were obtained via the minimization of a cost function, through a non-linear least-squares-fit procedure performed on the difference between experimental data and the computational results [32,39]. Material stability was also checked according to Drucker Stability Postulate [36]. The mean data of equilibrium stress–strain and stress relaxation–time curves were evaluated considering the four plantar foot regions separately. First, the hyperelastic parameters were identified considering the stress–strain response at equilibrium. The stress–strain analyses were obtained in the model by compressing the sample up to 50% strain. Then, the viscous parameters were obtained considering the stress–relaxation results. To minimize the number of parameters and correctly interpret the trend of the experimental data, two viscous branches were assumed. At last, the stress–strain curves, obtained from the unconfined compression tests performed at the three different strain rates, were used to validate the finite element model. In this case, three different analyses were performed, one for each strain rate, by compressing the sample up to 50% strain with a strain rate of 7%/s, 70%/s, and 700%/s.

An additional validation was performed considering indentation tests. FE models were developed in order to interpret the indentation protocol, designing a spherical indenter (represented as a sphere with a 15 mm diameter) and a cylindrical shaped sample (20 mm diameter, 10 mm height). The indenter was assumed to be a rigid body, while a friction coefficient equal to 0.1 was imposed to describe the interface between the indenter and the tissues [37]. Specific boundary conditions were applied to the sample in order to mimic the role of the cylinder containing it during the indentation experimental tests. The curved surface of the sample was fixed along the x-axis and the z-axis while an interlocking constraint was applied to the bottom area. The indentation procedure was mimicked by applying a vertical displacement, along the y-axis, to the indenter sphere towards the sample model. The discretization of finite elements consisted of linear 8-node hexahedral elements (C3D8RH). A sensitivity analysis of the mesh discretization was performed by varying the element size and comparing the results. The best compromise between result accuracy and computational burden was found by imposing an average mesh seed of 0.65 mm, which led to a 23,000-node model. The simulations consisted of an indentation that reached 50% deformation.

### 2.6. Statistical Analyses

Normality of the data distribution was checked using the Shapiro–Wilk test. One-way ANOVA with Tukey’s post hoc or Kruskal–Wallis with Dunn’s post hoc test were used to compare continuous variables depending on the distribution of the data.

The initial and final elastic modulus and relative stiffness were reported using box and whisker plots; the box shows the median and interquartile range (IQR 25th–75th percentiles), while whiskers report the highest and lowest value.

For all the analyses, a *p*-value less than 0.05 was considered statistically significant.

The statistical analysis was performed using GraphPad Prism 9.5.1.733 for Windows, GraphPad Software, Boston, MA, USA, www.graphpad.com (accessed on 10 November 2023).

## 3. Results

The post-processing of the stress relaxation experimental activities allowed us to achieve stress–strain at equilibrium and normalized stress–time curves from the unconfined compression tests; similarly, force–indentation strain at equilibrium and normalized force–time curves were obtained from the indentation procedures. These results showed the typical behaviour of adipose tissues, which is the non-linear and time-dependent response, as shown in Figure 4a and Figure 5a and in Figure 4b and Figure 5b, respectively. The strain was limited to the 50% value both in the stress–strain and force–strain curves, obtained from compression and indentation tests. Indeed, the real maximum strain reached by the plantar adipose tissue under compression, obtained through in vivo measurement of the heel pad thickness at 100% loading conditions, is around 50% [15,40,41], and beyond this value, the tissue deteriorates. The initial and final elastic moduli of the stress–strain equilibrium curves were calculated for each foot region (Figure 4c,d), as well as initial and final indentation stiffnesses of the force–indentation strain curves (Table 2).

Observing the initial elastic modulus, all regions showed similar values, while they were statistically different considering the final elastic moduli. More precisely, the HP, L, and M regions differ from the Met one (*p* < 0.006). The drop of normalized stress with time (1-*γ*∞) was computed for each foot region, highlighting a value around 75% for the Met, HP, and L regions, while a higher value was computed for the M ones, 87% (Figure 4e). The plateau reached at the end of 300s, the micro-structural rearrangement phenomena, due to the flux of liquid components and the alignment of the fibres, developed almost completely. Considering the indentation stiffness, reported in Table 2, no differences were found comparing the initial and the final values among the foot regions investigated except for the HP vs. Met initial ones (*p* = 0.022).

Regarding the unconfined compression tests at different strain rates, stress–strain curves were analysed at 7%/s, 70%/s, and 700%/s for each foot location. First of all, each region was investigated at all three different strain rates; then, all regions were compared among them considering a specific velocity (Figure 6). Also, in these cases, the initial and final elastic moduli were computed and are reported in Table 3 and Table 4.

Observing Figure 7, it is possible to appreciate that the results showed differences, comparing the initial elastic modulus, between the HP, M, and Met regions at 7%/s strain rates (*p* < 0.03), while no statistical differences were observed analysing data at the other two higher strain rates (70%/s and 700%/s). In addition, considering the initial modulus within a specific region, it is possible to appreciate significant statistical differences between the values obtained for the three strain rates (*p* < 0.0001 for HP, *p* < 0.0009 for L, *p* < 0.016 for M, and *p* < 0.009 for Met). Regarding the final elastic modulus, no statistical differences are highlighted, neither when comparing the foot regions at a specific strain rate or evaluating the modulus within a specific foot region at the three different strain rates.

To implement the plantar adipose tissue in the computational model of the foot, an almost incompressible (v = 0.499 [42]), isotropic, visco-hyperelastic model with two viscous branches was adopted. The constitutive model, with its parameters (Table 5), was able to interpret the behaviour of the adipose tissue in the different foot regions and during different types of tests, in terms of the non-linear (Figure 8a unconfined compression and Figure 8c indentation tests) and time-dependent response (Figure 8b). The contours of the minimum principal strain and stress, reached at the end of a computational analysis simulating an unconfined compression test, are reported in Figure 8d,e, allowing us to evaluate the strain and stress distribution occurring inside the adipose sample during compression. Instead, an example of minimum principal strain contour, achieved during indentation tests, is reported in Figure 8f. To evaluate the discrepancy between the experimental data and the model results, the mean absolute percentage error procedure, in terms of loads, was adopted. Concerning the equilibrium stress–strain curves obtained through unconfined compression tests, the value of discrepancy computed for each foot region is 8.23% for the HP, 11.34% for the L, 8.54% for the M, and 4.61% for the Met. The computational stress–strain at different strain rates and force–indentation strain results fall inside the mean curve ± Standard Deviation. In the first case, the percentage error values range between 55 and 200%, while they are about 243.99% for the HP, 303.58% for the L, and 35.85% for the Met region for the indentation force–indentation strain curves.

## 4. Discussion

To the best of the authors’ knowledge, this is one of the first studies in which multiple types of experimental tests were performed on samples from different plantar regions and then used to determine, through an inverse FEA, the mechanical parameters of the isotropic, incompressible, visco-hyperelastic constitutive model (adopted to describe the plantar adipose tissue mechanical behaviour).

After the above-mentioned procedure, the attention was focused on highlighting similarities and differences in the mechanical response of the plantar tissue within the four foot regions considered.

Regarding both the stress–relaxation and the unconfined compression tests performed at different strain rates, sample preconditioning was applied. According to other studies found in the literature, it is preferable to perform sample preconditioning when investigating biological tissue properties [43]. Indeed, it was observed that this procedure contributes in obtaining repeatable and comparable results, thus reducing experimental variability, even if samples are then tested at different strains [44].

Stress–relaxation tests allowed assessing that the region that exhibits a markedly different viscoelastic behaviour is the medial one. First of all, it has a drop of normalized stress with time that is about 87%, higher than the other locations; then, it presents a final elastic modulus at equilibrium that is 1.7 times higher than those of the HP and L and about 8 times higher than the elastic modulus of the Met region. These data demonstrate that differences of mechanical properties between the medial region and the others are not negligible, even if they were not experimentally investigated and considered until now. On the contrary, the other foot regions (HP, L, and Met) do not show a considerable different behaviour, as demonstrated by the initial and final elastic modulus and initial and final indentation stiffness computed from the unconfined compression tests and indentation ones, respectively.

Considering the different strain rates adopted in the unconfined compression tests, except for the initial elastic modulus computed for the lower velocity (7%/s), no significant differences were found between the foot regions, both for the initial and final elastic modulus. Also in this case, the medial region is the stiffest region. However, it is interesting to observe that each region, stressed with an increasing strain rate, exhibits an increasing initial elastic modulus, while the final one increases from lower to middle velocities (from 7%/s to 70%/s), but then unexpectedly decreases when tested with the highest strain rate (700%/s). It is probable that this last aspect may be due to the experimental protocol adopted and machine settings.

All the data and mechanical parameters obtained in this study suggest that the plantar adipose tissue exhibits an incompressible, isotropic, non-linear stress–strain characteristic and time-dependent behaviour. Moreover, it is possible to order the foot regions considering their stiffness, M > HP > L > Met, highlighting an agreement with previous works [23] (that consider only HP, L, and Met regions) and a correspondence between this order and the ones of the volume fraction of fibrous septa of the corresponding region [45].

The constitutive formulation adopted to describe the plantar adipose tissue mechanical behaviour (an incompressible, isotropic, visco-hyperelastic model) agrees with studies in which other adipose tissues were considered (e.g., infrapatellar and abdominal) [16,31,32,46]. The constitutive parameters were identified using the inverse analysis of the experimental tests, and the good agreement between experimental results and the model defines the ability of the proposed formulation to interpret the mechanical response of the plantar soft tissue. Moreover, the numerical analyses performed at the same differing strain rates of those adopted for the unconfined compression tests (Figure 8b) validate the constitutive model and the identified parameters.

Over the years, different studies investigated the plantar adipose tissue behaviour, through in vitro tests [22,23,27] or using in vivo procedures [47,48], and then defined a constitutive model to describe the tissue properties. Almost all the previous work, which adopted the in vitro procedures, performed tests on cadaveric feet, while the present work considers both human donors and cadaveric ones. The protocols adopted for the in vitro and in vivo experiments are different, both in terms of testing set-up, parameters, and sample configuration. All these aspects make the comparison between the results of the different works difficult. Especially in an in vivo test, it is difficult to isolate the contribution of only adipose tissue; indeed, the results are affected also by the other foot tissue mechanics. Anyway, it is possible to perform some comparison of the stress–strain response of the heel pad tissue, obtained through the unconfined compression tests at the lower strain rate (7%/s, Figure 6a), with other studies in which it was investigated both with in vivo and in vitro procedures at a similar strain rate. The present curve was found to be situated lower than stress–strain curves achieved through in vivo indentation [49,50,51] tests and a little bit higher than in the indentation in vitro procedure achieved on a cadaveric sample [52]. One of the possible reasons for this difference may be due to the fact that in in vivo tests, subjects were younger than those considered in this study. Another possible explanation is that the samples tested here consisted only of adipose tissue.

Considering the constitutive formulation, different literature works aimed at characterizing the plantar adipose tissue, but it is difficult to find a work in which the incompressibility, non-linearity, and time dependency are all together contemplated. Indeed, only one work adopted a visco-hyperelastic model and in vitro tests, as in the present study [22], but was limited to the heel pad region. Other works formulated a quasi-linear viscoelastic model (QLV, described by Fung [28]) to describe the mechanical behaviour of the different foot regions, but without including the hyperelastic phenomena [23,24,27]. Referring to in vivo experimental data, authors used a hyperelastic [48] or a visco-hyperelastic model [47] to describe the adipose tissue behaviour but obtained different parameters if compared to this work. This is principally due to the fact that with the in vivo test, it is difficult to isolate the contribution of only adipose tissue; indeed, the results are affected also by the other foot tissue mechanics. As an example, the shear modulus computed for the plantar adipose tissue and reported in Kwak et al. (2020) [48] is about four orders of magnitude bigger than those reported in the present Table 2.

Comparing plantar adipose tissue behaviour with adipose tissues from other body regions, it is possible to appreciate that it is less stiff than the Infrapatellar Fat Pad (IFP) [31] and the abdominal adipose tissue [32]. In particular, the initial stiffness is about half the values of those of the IFP and the abdominal adipose tissue, while the final ones are a little bit lower than the infrapatellar and abdominal adipose tissues but of the same order of magnitude.

Unfortunately, even though the plantar adipose tissue is well described and characterized in all its regions with different types of tests here, including a complete constitutive model, some limitations should be mentioned. First, experimental tests were performed at room temperature, which is about 25 °C, far from the body one. Secondly, the number of patients involved in the study was limited (*n* = 7) and for this reason it was not possible to compare the experimental results considering subjects’ parameters, e.g., the age or the BMI. In addition, samples were harvested only from healthy feet of middle-aged patients, preventing the possibility of comparing their mechanical properties with diseased tissues, such as diabetic [26,27,48] or osteoarthritic tissues, and with younger or older subjects [22,53]. Finally, experimental tests were carried out only on adipose samples, not considering, for instance, the behaviour of the structure formed by adipose tissue and the underlying skin.

A future step will be to overcome the limitations just mentioned. One of the first improvements will be to test samples at higher temperatures, like 35–37 °C [23,27], to emulate the in vivo conditions. Then, many patients of a wide range of ages will be involved in the study to also assess diseased adipose tissue properties. In the end, the goal will be to perform experimental tests (e.g., indentation ones) on samples formed by adipose tissue and the underlying skin, to evaluate the mechanical behaviour of the whole structure.

## 5. Conclusions

The work aimed at characterizing the mechanical behaviour of the plantar adipose tissue, with respect to the different foot regions. Thanks to experimental results, the constitutive parameters’ model identification has been performed and then included in the computational model of the foot. The numerical model may provide information on mechanical behaviour of plantar tissues during functional loading conditions, but it can also be a valuable tool to simulate different situations, such as the interaction between the foot and footwear, or to compare the mechanical response of tissues in healthy and pathologic conditions. All these pieces of information will be useful tools to improve computational models of the foot, able to mimic its real mechanical behaviour, thus providing new insights in the medical and bioengineering area, such as in the designing of, e.g., specific insoles for diabetic patients to minimize pain caused by plantar ulcers.

## Figures and Tables

**Figure 1 bioengineering-11-00042-f001:**
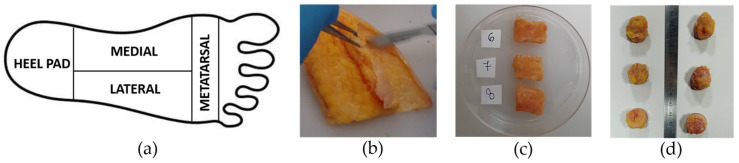
(**a**) Foot subdivision into heel pad (HP), medial (M), lateral (L), and metatarsal (Met) regions, (**b**) dissection of adipose tissue from the skin, (**c**) preparation of cubic plantar adipose samples, and (**d**) cylindric samples for unconfined compression and indentation tests, respectively.

**Figure 2 bioengineering-11-00042-f002:**
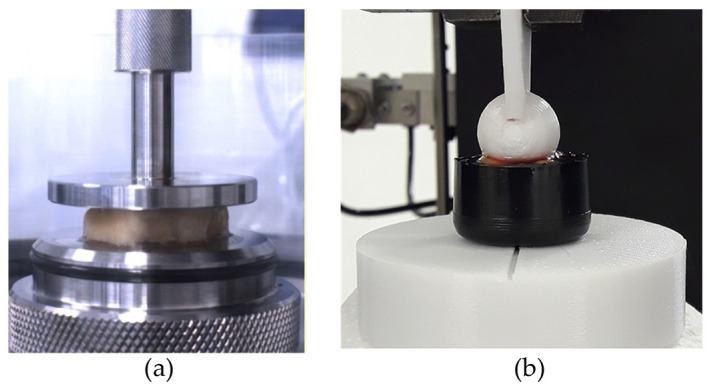
(**a**) Example of unconfined compression test and (**b**) indentation test.

**Figure 3 bioengineering-11-00042-f003:**
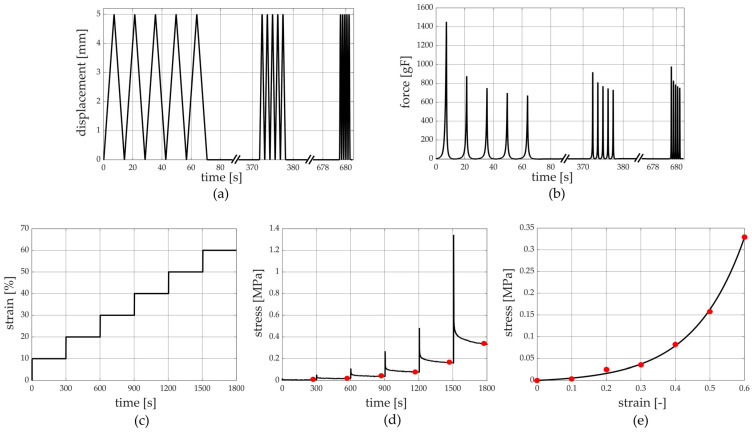
Unconfined compression tests at different strain rates’ protocol: (**a**) imposed displacement vs. time and (**b**) the measured force vs. time. Stress relaxation protocol: (**c**) imposed compressive strain history vs. time and (**d**) stress vs. time with evaluation of equilibrium conditions and (**e**) stress vs. strain at equilibrium.

**Figure 4 bioengineering-11-00042-f004:**
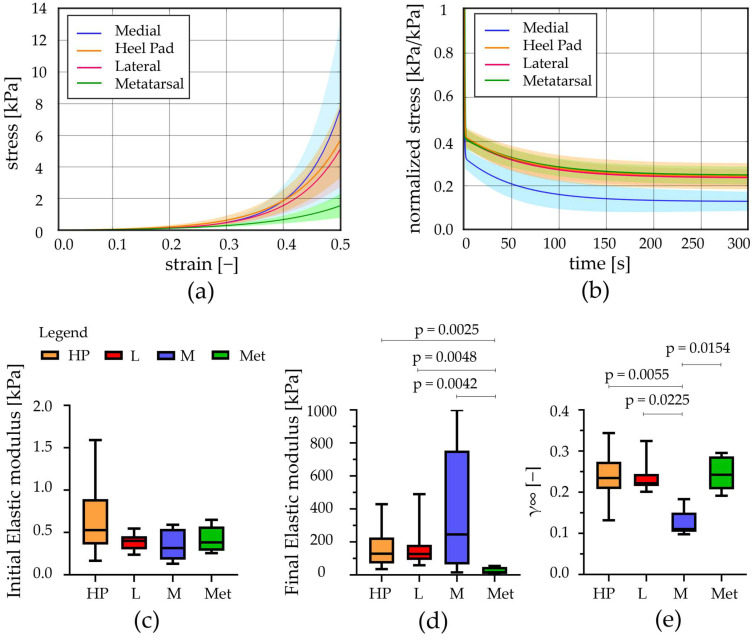
Unconfined compression tests: (**a**) Stress–strain equilibrium curves and (**b**) normalized stress–time results comparing the different regions (mean ± SD). (**c**) Initial and (**d**) final elastic modulus computed from stress–strain equilibrium curves, (**e**) relative stiffness computed at t = 300 s (median with maximum and minimum values). HP = heel pad; L = lateral; M = medial; Met = metatarsal; *p* = statistically different *p*-value.

**Figure 5 bioengineering-11-00042-f005:**
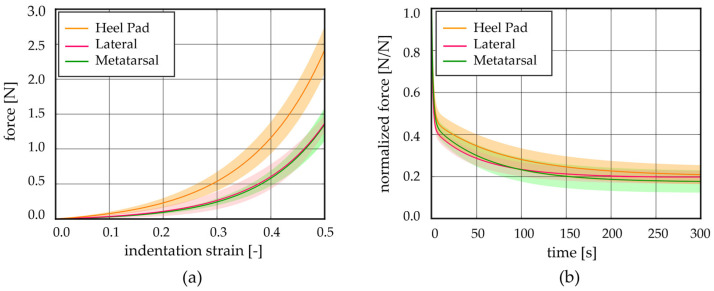
Indentation tests: (**a**) force–indentation strain equilibrium curves and (**b**) normalized force–time results comparing the different regions (mean ± SD).

**Figure 6 bioengineering-11-00042-f006:**
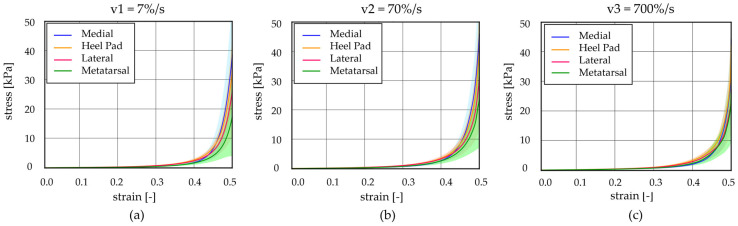
Unconfined compression tests at different strain rates: stress–strain curves at (**a**) v1 = 7%/s, (**b**) v2 = 70%/s, and (**c**) v3 = 700%/s comparing the heel pad, medial, lateral, and metatarsal regions (mean ± 90% CI).

**Figure 7 bioengineering-11-00042-f007:**
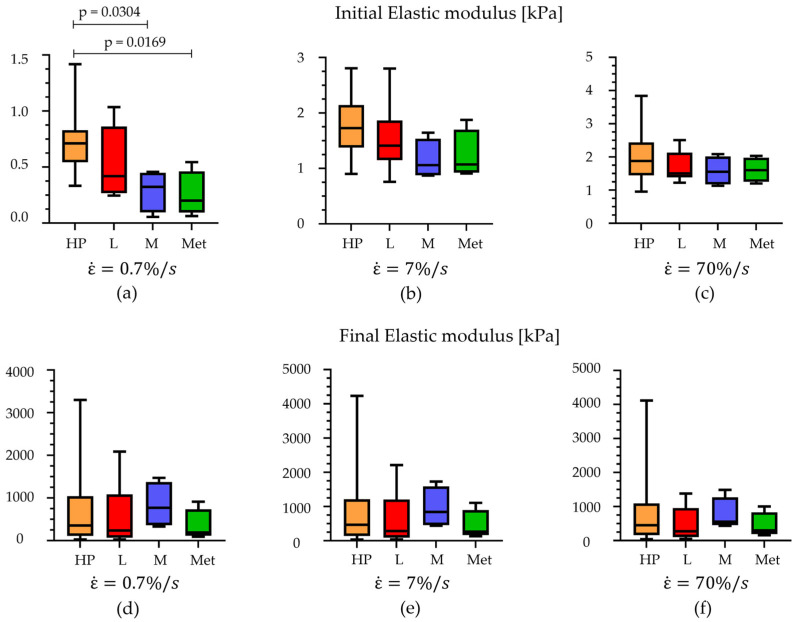
Comparison of the initial elastic modulus at (**a**) 7%/s, (**b**) 70%/s, and (**c**) 700%/s of strain rates and comparison of the final elastic modulus at (**d**) 7%/s, (**e**) 70%/s, and (**f**) 700%/s within the foot regions (median with maximum and minimum values). HP = heel pad; L = lateral; M = medial; Met = metatarsal.

**Figure 8 bioengineering-11-00042-f008:**
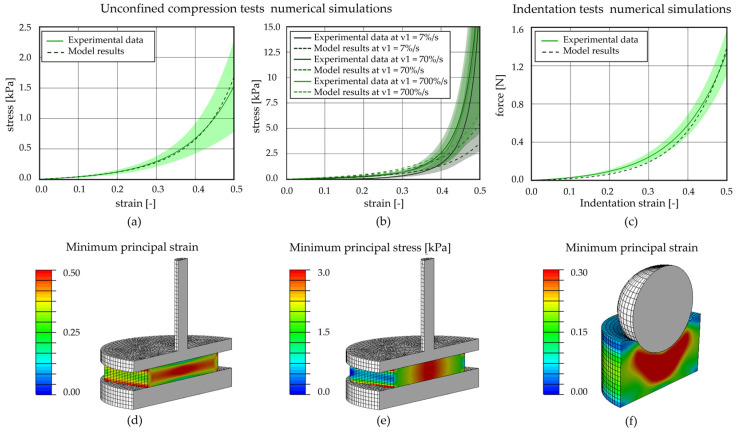
Computational analysis of unconfined compression and indentation tests for the metatarsal region. Unconfined compression (**a**) stress–strain equilibrium curve and (**b**) stress–strain curves at three different strain rates; (**c**) indentation force–strain equilibrium curve. Contours of (**d**) minimum principal strain and (**e**) minimum principal stress at the applied 50% strain reached at the end of the compression procedure; (**f**) contours of minimum principal strain at the applied 30% strain of the indentation tests.

**Table 1 bioengineering-11-00042-t001:** Patients’ data.

Subject	Sex	Age (Years)	BMI (kg/m^2^)	Pathology	Amputation Site, Leg	a/c
1	M	40	25.3	Chondrosarcoma	Interileoabdominal, RL	a
2	M	49	28.7	Chondrosarcoma	Interileoabdominal, LL	a
3	M	54	23.7	Chondrosarcoma	Thigh, RL	a
4	M	82	26.7	Leiomyosarcoma	Thigh, RL	a
5	F	58	24.8	Synovial sarcoma	Thigh, RL	a
6	M	67	26.1	-	RL and LL	c
7	F	62	25.7	-	RL and LL	c
Mean ± SD		59 ± 13	25.9 ± 1.6			

Sex: M = male, F = female, Age = age at the amputation; LL = left leg, RL = right leg; a = alive, c = cadaver.

**Table 2 bioengineering-11-00042-t002:** Initial and final indentation stiffness computed from force–indentation strain equilibrium curves; relative stiffness computed from normalized force–time curves at t = 300 s.

Region	Initial Indentation Stiffness (N)	Final Indentation Stiffness (N)	Relative Stiffness *γ*∞ (−)
HP	0.82 ± 0.27 *	16.34 ± 1.01	0.20 ± 0.04
L	0.38 ± 0.27	10.29 ± 1.57	0.20 ± 0.03
Met	0.31 ± 0.07 *	9.60 ± 1.54	0.17 ± 0.05

HP = heel pad; L = lateral; Met = metatarsal. Data are shown as mean value ± Standard Deviation; *: HP vs. Met, *p*-value = 0.022.

**Table 3 bioengineering-11-00042-t003:** Initial elastic modulus for the foot regions at three different strain rates.

Initial Elastic Modulus (kPa)				
Region	${\dot{\varepsilon }}_{1}$ = 7%/s	${\dot{\varepsilon }}_{2}$ = 70%/s	${\dot{\varepsilon }}_{3}$ = 700%/s	*p*-Value
HP	0.73 ± 0.27	1.78 ± 0.53	1.97 ± 0.71	a: *p* < 0.0001; b: *p* < 0.0001
L	0.53 ± 0.31	1.55 ± 0.62	1.70 ± 0.46	a: *p* < 0.0002; b: *p* < 0.0009
M	0.29 ± 0.19	1.16 ± 0.36	1.58 ± 0.44	a: *p* < 0.0160; b: *p* < 0.0014
Met	0.25 ± 0.21	1.23 ± 0.44	1.61 ± 0.38	a: *p* < 0.0094; b: *p* < 0.0012

HP = heel pad; L = lateral; M = medial; Met = metatarsal. Data are shown as mean value ± Standard Deviation. a: 
${\dot{\varepsilon }}_{1}$
 vs. 
${\dot{\varepsilon }}_{2}$
; b: 
${\dot{\varepsilon }}_{1}$
 vs. 
${\dot{\varepsilon }}_{3}$
; c: 
${\dot{\varepsilon }}_{2}$
 vs. 
${\dot{\varepsilon }}_{3}$
.

**Table 4 bioengineering-11-00042-t004:** Final elastic modulus for the foot regions at three different strain rates.

Final Elastic Modulus (kPa)				
Region	${\dot{\varepsilon }}_{1}$ = 7%/s	${\dot{\varepsilon }}_{2}$ = 70%/s	${\dot{\varepsilon }}_{3}$ = 700%/s	*p*-Value
HP	690.88 ± 853.66	818.04 ± 1057.22	755.98 ± 1007.78	>0.05
L	551.82 ± 757.93	587.05 ± 803.64	441.53 ± 514.83	>0.05
M	833.79 ± 538.24	944.67 ± 609.28	769.95 ± 493.42	>0.05
Met	352.91 ± 379.68	442.41 ± 448.06	437.37 ± 381.05	>0.05

HP = heel pad; L = lateral; M = medial; Met = metatarsal. Data are shown as mean value ± Standard Deviation.

**Table 5 bioengineering-11-00042-t005:** Visco-hyperelastic constitutive parameters for adipose tissue of different foot regions.

Region	µ (kPa)	*α* (−)	*D* (kPa^−1^)	*γ*_1_ (−)	*γ*_2_ (−)	*τ*_1_ (s)	*τ*_2_ (s)
HP	0.14	−6.18	0.03	0.58	0.18	0.39	63.48
L	0.08	−7.01	0.05	0.58	0.18	0.37	62.87
M	0.07	−7.95	0.06	0.67	0.20	0.39	54.44
Met	0.10	−4.09	0.04	0.58	0.17	0.36	66.50

HP = heel pad; L = lateral; M = medial; Met = metatarsal.

## Data Availability

Data are available, contacting the corresponding authors, on reasonable request.

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
