# Peer review of "Mechanical Behaviour of Plantar Adipose Tissue: From Experimental Tests to Constitutive Analysis"

_bioengineering, 2023, doi:10.3390/bioengineering11010042_

Round 1

Reviewer 1 Report

Comments and Suggestions for Authors

Mechanical behaviour of plantar adipose tissue: from experimental tests to constitutive analysis

-----------

The paper identifies a constitutive model for plantar adipose tissue based on experiments. The tissue is extracted from 4 different areas of the plantar area from both donors and cadavers. The constitutive models are: nonlinear hyperelastic (Ogden model) for the elastic part, and a viscoelastic model for the rate-dependent part.  In contrast with other papers in the literature, samples are taken not only from cadavers but also from donors, and includes the medial region of the foot.

The thematic is interesting, the paper is well written, and results are well documented and explained.

Comments:

- In this study, samples from donors and cadavers are considered together. Do you think that you may find significative differences in comparing these tissues separately?

- As the authors acknowledge in the discussion, 7 samples seems to be at the limit for obtaining significant statistical data, specially when no distinction has been made with age and BMI.

- What is the justification if using the Ogden model, among many hyperelastic models of the literature?

- The viscoelastic model is kinematically nonlinear, since accounts for large strains, but the stress-strain history comes from a linear constitutive equation for the stress-like viscous variables. This make the model suitable for small strain rates.  Are the strain rates of the experiments in the range of applicability of the model?

- Could you provide some details about the cost function employed in the inverse FEA? It is just said that "consists of the difference between experimental data and the computational results" and provide reference [34]. But in [34] no further details are found.

- The finite elements employed in ABAQUS are reported to be linear hexahedral elements. Could you provide the exact type? Be aware that isoparametric linear hexahedrals do not behave well in cuasi-incompressible problems, as the one you are considering (nu=0.499).

- Fig 8b), it looks that the difference between experiment and numerical model for large strains is very large. I am not sure that the model is working for this situation.

Comments on the Quality of English Language

English language is correct, only some minor misspellings.

Reviewer 2 Report

Comments and Suggestions for Authors

The authors propose an extensive data set on the soft tissues of the foot. These data have been exploited by inverse analysis to propose hyper-viscoelastic modelling. The results constitute an important database.

The friction coefficient was chosen arbitrarily in the inverse analyses, so what impact does it have on the identification of behavior law parameters? Are the results of the inverve analysis modified?

The use of an Ogden model in a hyperelastic setting requires the mu alpha product to be positive. Is this condition lifted when the model is used in a viscoelastic setting and the other parts of the model contribute? Can the authors theoretically justify these parameter values?

Is the value of the initial modulus really reliable, since it is always very difficult to determine the start of tests with non-perfect specimen geometries such as those of biological tissues? The deviation seems to me to be extremely small for a large number of potential uncertainties.

Reviewer 3 Report

Comments and Suggestions for Authors

It is a novel study and can be accepted. However, minor corrections should be made before it can be published. See my comments below.

1.     Abstract

a)     Include a solid conclusion of the study.

2.     Introduction

a)     Research gap – it is not clear what is the research gap from the introduction part. What is the main problem of the previous study on this issue? What are the drawbacks and limitations of the previous literature? Need to clearly state here to make sure that the current manuscript addresses all the previous problems and from this can indicate the novelty of the current study.

3.     Materials and Methods

a)     Suggest including consent forms in the appendix.

b)    Line 96 – found the different font type.

c)     State the load rate of the machine during the compression testing. Is there any standard (ASTM or ISO) that authors refer it? Please include.

d)    The jig for both compression and indentation tests should be clearly stated here.

e)     How many samples for the mechanical testing? It is a destructive testing, so authors should prepare at least 3 for each subject. Please clearly explain this.

f)     Mesh convergence study or mesh sensitivity study. This is also not reported as a figure. So how do authors decide the best elements' size or numbers? Please include a figure of the results.

g)    Please explain further the use of hexahedral elements in this case. Any references? Since the use of tetrahedral is one of the options to simplify the method. Discuss it in the discussion section.

h)    Explain further about the geometry of the model in FEA.

4.     Results

a)     No issue.

5.     Discussion

a)     Very limited discussion about the results and comparison between the author’s results with previously published literature by others. Authors should compare, at least find the difference between the stress from others and their own.

b)    Should include more limitations of the study. What can be improved in the future? Suggest a minimum of 2 paragraphs.

6.     Reference

a)     Need to add more references. Especially for the discussion part.

Comments on the Quality of English Language

Minor errors in some sentences. Please check all.

Round 2

Reviewer 2 Report

Comments and Suggestions for Authors

Thank you for all the corrections.

Nevertheless, I still disagree on the problem of stability. In his paper Ogden (Fitting hyperelastic models to experimental data - Computational Mechanics 34 (2004) 484–502 -DOI 10.1007/s00466-004-0593-y ) eq (22), Ogden explained that alpha * mu must be positive. In my opinion, the parameters need to be re-identified to verify this condition.
